# Intestinal Microbiota and Derived Metabolites in Myocardial Fibrosis and Postoperative Atrial Fibrillation

**DOI:** 10.3390/ijms25116037

**Published:** 2024-05-30

**Authors:** Antonio Nenna, Alice Laudisio, Chiara Taffon, Marta Fogolari, Cristiano Spadaccio, Chiara Ferrisi, Francesco Loreni, Omar Giacinto, Ciro Mastroianni, Raffaele Barbato, David Rose, Antonio Salsano, Francesco Santini, Silvia Angeletti, Anna Crescenzi, Raffaele Antonelli Incalzi, Massimo Chello, Mario Lusini

**Affiliations:** 1Cardiac Surgery, Fondazione Policlinico Universitario Campus Bio-Medico, Via Alvaro del Portillo 200, 00128 Rome, Italy; chiara.ferrisi@unicampus.it (C.F.); francesco.loreni@unicampus.it (F.L.); o.giacinto@policlinicocampus.it (O.G.); c.mastroianni@policlinicocampus.it (C.M.); r.barbato@policlinicocampus.it (R.B.); m.chello@policlinicocampus.it (M.C.); m.lusini@policlinicocampus.it (M.L.); 2Internal Medicine, Fondazione Policlinico Universitario Campus Bio-Medico, 00128 Rome, Italy; a.laudisio@policlinicocampus.it (A.L.); r.antonelli@policlinicocampus.it (R.A.I.); 3Pathology, Fondazione Policlinico Universitario Campus Bio-Medico, 00128 Rome, Italy; c.taffon@policlinicocampus.it (C.T.); a.crescenzi@policlinicocampus.it (A.C.); 4Clinical Laboratory, Fondazione Policlinico Universitario Campus Bio-Medico, 00128 Rome, Italy; m.fogolari@policlinicocampus.it (M.F.); s.angeletti@policlinicocampus.it (S.A.); 5Cardiac Surgery, University of Cincinnati Medical Center, Cincinnati, OH 45219, USA; cristianospadaccio@gmail.com; 6Cardiothoracic Surgery, Lancashire Cardiac Centre, Blackpool Teaching Hospital, Blackpool FY3 8NP, UK; mr.rose@nhs.net; 7Cardiac Surgery, Ospedale Policlinico San Martino, University of Genoa, 16126 Genoa, Italy; ant.salsano@gmail.com (A.S.); francesco.santini@unige.it (F.S.)

**Keywords:** microbiome, microbiota, atrial fibrillation, cardiac surgery, fibrosis

## Abstract

The high incidence of atrial fibrillation (AFib) following cardiac surgery (postoperative atrial fibrillation, POAF) relies on specific surgical features. However, in the setting of POAF, the role of the microbiome in the modulation of cardiac fibrosis is still not clear. This study aimed to analyze the effect of the microbiome and its main metabolic product (trimethylamine-N-oxide, TMAO) in the fibrosis of myocardial tissue, to investigate its role in POAF. Patients undergoing elective cardiac surgery with cardiopulmonary bypass, central atrio-caval cannulation and no history of AFib, were included. A fragment of the right atrium was analyzed for qualitative and mRNA-quantitative evaluation. A preoperative blood sample was analyzed with enzyme-linked immunosorbent assay (ELISA). A total of 100 patients have been included, with POAF occurring in 38%. Histologically, a higher degree of fibrosis, angiogenesis and inflammation has been observed in POAF. Quantitative evaluation showed increased mRNA expression of collagen-1, collagen-3, fibronectin, and transforming growth factor beta (TGFb) in the POAF group. ELISA analysis showed higher levels of TMAO, lipopolysaccharide and TGFb in POAF, with similar levels of sP-selectin and zonulin. TMAO ≥ 61.8 ng/mL (odds ratio, OR 2.88 [1.35–6.16], *p* = 0.006), preoperative hemoglobin < 13.1 g/dL (OR 2.37 [1.07–5.24], *p* = 0.033) and impaired right ventricular function (OR 2.38 [1.17–4.83], *p* = 0.017) were independent predictors of POAF. Also, TMAO was significantly associated with POAF by means of increased fibrosis. Gut microbiome product TMAO is crucial for myocardial fibrosis, which is a key factor for POAF. Patients in preoperative sinus rhythm who will develop POAF have increased genetic expression of pro-fibrotic genes and enhanced fibrosis in histological staining. Elevated TMAO level (≥61.8 ng/mL) is an independent risk factor for POAF.

## 1. Introduction

Atrial fibrillation (AFib) is a very common arrythmia in the general population, with an incidence of 1–2% [1,2,3]. The consequences of atrial fibrillation are correlated to an embolic risk and a reduced ejection fraction of the ventricle due to a rapid ventricular response maintained over time (called “tachy-cardiomyopathy”); therapies based on anticoagulant and antiarrythmia drugs might increase the risk of hemorrhagic or bradycardic complications, respectively [1]. The etiopathogenesis depends on several factors affecting the cardiac electrical pathways, such as delayed transvers propagation, unidirectional conduction block and reentry, hetero-cellular interactions that enhanced automaticity and paracrine action [1,2,3,4].

The risk factors of AFib may induce modifications of cardiac cells and myocardial interstitium leading to atrial fibrosis [2,3,4]. The main pathways involved in fibrotic processes depend on transforming growth factor beta (TGFb)/small mothers against decapentaplegic (SMAD) and mitogen-activated protein kinase (MAPK) to produce increased deposition of collagen, fibronectin and extracellular matrix protein [2,3,4,5]. The recent literature analyzed the role of the gut microbiota in these mechanisms. In animal models, the trimethylamine-N-oxide (TMAO) is involved in myocardial fibrosis, thanks to the activation of an inflammasome through the “nucleotide-binding oligomerization domain, leucine rich repeat and pyrin domain containing” (NLRP3) pathway [6], and TMAO may induce hypertrophy and fibrosis via TGFb/SMAD-2 pathways [7,8].

The analysis of microbiota and its metabolites have been identified as the main conditions for increased mortality and poor prognosis in patients with cardiovascular disease [9]. In animal models, age-related intestinal dysbiosis promotes atrial fibrillation through increased levels of lipopolysaccharide and glucose, with the activation of the NLRP3 inflammasome, leading to atrial tissue fibrosis. Microbiota transplantation in animal models, or selective inhibition of NLRP3, reduced the fibrosis process and the arrhythmia susceptibility, underlining the role of systemic inflammation related to dysbiosis as a pathogenetic element of atrial fibrillation [10,11,12]. The recent literature focused on the role of TMAO and the effects of microbiota in atrial fibrillation in general population [10,13,14]. Several bacterial groups of the gut microbiota have been related to different patterns of atrial fibrillation (paroxysmal or persistent) and the time of events of atrial fibrillation [13]. The role of TMAO has been underlined as a final pathway by which metabolism of the microbiota leads to clinical effects [12,15,16,17]. A recent meta-analysis showed a strong correlation between atrial fibrillation and serum TMAO levels in the general population, with the risk of having atrial fibrillation increasing from 6% for each 1 µmol/L increment to 73% for each 10 µmol/L increment (odds ratio, OR 1.73, 95% CI 1.05–2.86) [18].

Postoperative atrial fibrillation (POAF) is the most common arrythmia after cardiac surgery with an incidence of 30–35% (30 times greater than “common” AFib) and it is associated with detrimental short term outcomes such as a higher risk of stroke (about 3 times greater than general population), and long term outcomes such as a significant increase in mortality rate [19,20]. The greater incidence of POAF compared to “common” AFib depends on specific surgical features, such as atrial manipulation/atriotomy/cannulation for cardiopulmonary bypass, pericardiectomy with increase in intrapericardial inflammatory cytokines (called “sterile inflammation”), perioperative autonomic demodulation, inflammation and oxidative stress [20].

Over the years, many factors have been shown to be involved in atrial remodeling and POAF [1], but the role of microbiome and its metabolites has not yet been analyzed in patients undergoing cardiac surgery. 

In this study, the authors aimed to identify risk factors associated with POAF through analyzing the role and effect of gut microbiota and its main metabolite (TMAO) on myocardial fibrosis (with histological study of the right atrium). Also, preoperative clinical or echocardiographic risk factors associated with POAF were identified.

## 2. Results

A total of 100 patients have been included, and POAF was observed in 38%. Results are shown based on primary endpoint (POAF versus sinus rhythm, “no POAF”). 

Preoperative characteristics are included in Table 1. The enrolled population has a high cardiovascular risk profile due to classic risk factors, with good left and right ejection fraction even if, in 44% of cases, patients present diastolic dysfunction. There are no significant differences in preoperative variables in patients who developed POAF; according to data in the literature, patients with POAF show a trend towards preoperative anemia (*p* = 0.062).

Based on the study’s inclusion and exclusion criteria, patients underwent cardiac surgery, in particular coronary artery bypass grafting (82%), aortic valve replacement (8%), or ascending aorta replacement (10%), using general anesthesia and cardiopulmonary bypass with single venous cannulation. Patients eligible for mitral or tricuspid valve surgery were excluded as the risk of postoperative atrial arrhythmias is closely related to cardiac remodeling due to pressure or volume overload. There are no significant differences in the overall operative risk of patients (EuroSCORE II) and in the duration of cardiopulmonary bypass and aortic clamping (Table 1).

Complications and postoperative characteristics are included in Table 2. POAF was the most frequent complication, and it was observed in 38% of patients. Pleural effusion drainage via thoracentesis was performed in 26% of cases, while the incidence of significant pericardial effusion (>1 cm or requiring surgical drainage) was 3%.

### 2.1. Histological Evaluation of the Right Atrium 

Histological analysis of the right atrial tissue was performed in all patients (Table 3). A higher degree of fibrosis was found in the atrial tissue of patients who developed atrial fibrillation (*p* = 0.001). Additionally, patients who developed POAF presented a higher level (qualitative) of angiogenesis and inflammatory cells in the right atrial tissue (Appendix A).

A greater histological degree of fibrosis had a 3.2 times higher incidence in patients who developed POAF. These data were confirmed through quantitative assessment using qRT-PCR (Table 4, Figure 1).

The qRT-PCR data showed an increase in gene expression for genes involved in fibrosis pathways (TGFb and SMAD-2) and extracellular matrix production (Collagen-1, Collagen-3, and Fibronectin) in patients who developed postoperative atrial fibrillation. Patients who develop POAF showed approximately a 20% increase in Collagen-1 expression, around a 25% increase in Collagen-3 and Fibronectin expression, and a 5–10% increase in TGFb and SMAD-2 expression, confirming the study hypothesis that extracellular matrix deposition is supported by TGFb/SMAD-related pathways.

### 2.2. ELISA Results

ELISA analyses performed on serum (with blood samples collected the evening before surgery) showed that patients who develop POAF had increased levels of TMAO (+28%, *p* = 0.001), TGFb (+32%, *p* = 0.001), and LPS (*p* = 0.001) (Table 5, Figure 2). Levels of zonulin and sP-selectin were similar between the two groups (*p* = 0.218 and *p* = 0.189, respectively) (Table 5, Figure 2).

### 2.3. Regression Analysis 

Preoperative and intraoperative clinical variables and serum values obtained from ELISA analysis were analyzed to identify predictive parameters of POAF through logistic regression analysis. Variables derived from histological analyses (both qualitative and quantitative) were excluded as their clinical application would be limited by the technical time required for the analyses. 

Results of the univariate and multivariate analyses are shown in Table 6. Variables with *p* < 0.200 were included in the initial multivariate model. This model was optimized using the backward stepwise approach, analyzing every single step with a likelihood ratio test and Wald test until reaching a unique model. The fitness of the final model was evaluated using c-statistic (AUC, area under the ROC curve), the Hosmer–Lemeshow test, the Link test, and partial residual plots for each variable (*logitcprplot*) (Appendix A). These analyses confirmed good discrimination and calibration of the model (AUC, c-statistic = 0.859 (95% CI 0.824–0.894); Hosmer–Lemeshow chi^2^ = 9.08, *p* = 0.335; Link test, prediction squared, *p* = 0.338) (Appendix A). 

In order to improve and simplify the clinical application of this regression model, a cut-off analysis based on AUC (*dtroc*) was performed for the 4 continuous variables included in the model to convert into binary variables. The cut off analysis is shown in Table 7. The regression model (Table 8) was adjusted using the obtained binary variables, and further model refinement was performed. 

Table 9 showed the final multivariate regression model with binary variables. Fitness analyses confirmed good discrimination and calibration of the model in Table 9 (AUC, c-statistic = 0.702 (95% CI 0.675–0.729; Hosmer–Lemeshow chi^2^ = 8.11, *p* = 0.230; Link test, prediction squared, *p* = 0.709). (Appendix A).

In conclusion, preoperative anemia (Hb < 13.1 g/dL, *p* = 0.033), reduced right ventricular systolic function (TAPSE < 20 mm, *p* = 0.017) and high serum TMAO levels (≥61.8 ng/mL, *p* = 0.006) are associated with a higher incidence of POAF (Table 9, Figure 3).

In order to investigate the correlation between the products of the gut microbiota (TMAO levels), presence of significant myocardial fibrosis, levels of fibrosis biomarkers, and risk of POAF, interaction analyses and regression with endogenous covariates (*ivprobit)* were performed. In a univariate regression, elevated TMAO levels (≥61.8 ng/mL) had an odds ratio of 2.67 (95% CI 1.19–5.98, *p* = 0.017) for myocardial fibrosis grade ≥ 2.

Considering POAF as the dependent variable, both TMAO (OR 4.16, *p* = 0.001) and fibrosis (OR 15.33, *p* = 0.001) were significantly linked in a univariate analysis. Then, the TMAO and fibrosis variables were evaluated both in bivariate analysis and through assessing their interaction. The effect of the interaction variable was found to be significant; therefore, TMAO and fibrosis are partially correlated (Table 10). In order to confirm the fibrosis-related association between TMAO and POAF, a probit regression model was analyzed with endogenous covariates, examining the relationship between histological fibrosis, biomarkers (TGFb in ELISA) and TMAO (Table 10). This model has an AUC (c-statistic) of 0.895 (95% CI 0.855–0.935).

### 2.4. Secondary Endpoints: Thoracentesis 

A sub-analysis of ELISA data was performed, analyzing the postoperative thoracentesis as an endpoint. Thoracentesis may be correlated with the electrolyte imbalance worsened by systemic inflammation from cardiopulmonary bypass and potentially intensified by changes in intestinal permeability. No statistically significant differences were shown, and no further descriptive analyses were performed.

## 3. Discussion

The aim of this study is to analyze the role of TMAO and myocardial fibrosis in atrial fibrillation pathogenesis, especially in postoperative atrial fibrillation, and the negative short and long term outcomes for patients that underwent cardiac surgery. This study was based on patients who underwent cardiac surgery in sinus rhythm. The histological analysis and blood samples were analyzed before atrial fibrillation occurred. The incidence of POAF was 38%, as data in the literature have shown. Patients with POAF showed high blood levels of TMAO, LPS and TGFb before cardiac surgery. Histological analysis of right atrium observed high levels of TGFb, highlighting the strong relationship with the TGFb/SMAD pathways and with the increase in collagen (col-1 and col-3) and fibronectin, as extracellular matrix proteins. The study recommended that intestinal microbiota disorders might be included in the model for predicting AFib recurrence. TMAO, as a metabolite derived from intestinal flora, may be associated with the inflammatory response and the activation of pro-inflammatory signals related to atrial fibrillation. Data have shown that high levels of TMAO (TMAO ≥ 61.8 ng/mL, OR 2.88 with *p* = 0.006) might increase the risk of POAF regardless of preoperative anemia (Hb < 13.1 g/dL, OR 2.37 with *p* = 0.033) and reduced right ejection fraction (TAPSE < 20 mm, OR 2.38 with *p* = 0.017).

POAF is the most frequent complication after cardiac surgery according to data in the literature [19,20], and in our study the incidence of POAF (38%) is similar to available evidence (30–35%) [19,20]. Several studies suggest that the gut microbiota might be closely linked to atrial fibrillation pathogenesis [21,22]. TMAO has been shown to increase autonomic activity and to stimulate the release of inflammatory cytokines, enhancing cardiac fibrosis thanks to the deposition of collagen by fibroblasts. In addition, TMAO might also promote AFib susceptibility due to pro-atherosclerotic mechanisms and cardiac remodeling through the activation of TLR-4-dependent pathways [21]. 

The concept of “gut–immune–heart” axis has been gradually introduced in the scientific literature, and the role of gut microbiome in atrial fibrillation has been extensively investigated in recent years [22]. Several mechanisms have been described in atrial fibrillation pathogenesis related to TMAO, including cerebrovascular diseases, in particular sympathetic nervous system, cardiovascular disease, cardiac remodeling, myocardial fibrosis, oxidative stress and endothelial dysfunction [23]. As preclinical and clinical studies in a cardiac surgery setting have shown, patients were more susceptible to the TMAO effects due to endothelial dysfunction and oxidative stress of cardiopulmonary bypass [21,23]. Elevated serum TMAO levels were predictive of thrombus formation in patients with atrial fibrillation, allowing for tailored thrombus-prevention strategies in selected patients with high susceptibility to thromboembolic complications in the general population (i.e., not undergoing cardiac surgery) [24]. The diagnostic potential of TMAO levels to identify thrombus formation was moderate (AUC 0.66, 95% CI 0.52–0.80, *p* = 0.01), but a clinically-relevant cutoff analysis was not performed by the authors [24].

“Classic” risk factors for POAF include preoperative, intraoperative and postoperative factors [20,25,26,27,28,29,30]. As for preoperative conditions, the strongest association is observed with chronic obstructive pulmonary disease, right coronary artery disease, anemia, enlargement of left atrium, reduced ventricular function and advanced age. Intraoperative factors include prolonged cross-clamp time and inotropic support. Postoperative factors include prolonged ventilation time and alterations in fluid balance. Our study confirmed the previous findings in terms of preoperative anemia and right ventricular dysfunction (that can be considered a manifestation of right coronary disease among patients with ischemic heart disease undergoing myocardial revascularization, as most of the included cohort). Interestingly, the magnitude of the effect related to TMAO was higher than “classic” risk factors. This might allow speculation that serum biomarkers might play a crucial role in the future, as they are independent from other clinical parameters that can be subjected to adjustments (e.g., preoperative transfusions) or intrinsic errors in measurements (e.g., operator-dependency).

In the specific setting of cardiac surgery, the only available study has been recently published by Wang et al. in 2023 [31]. They collected preoperative fecal samples from patients undergoing off-pump CABG and propensity score matched patients with or without POAF. Microbiome profile analysis was significantly different in patients with POAF (increase in Lachnospira, Acinetobacter, Veillonella and Aeromonas; decrease in Escherichia-Shigella, Klebsiella, Streptococcus, Brevundimonas and Citrobacter), concluding that gut composition was different in patients with preoperative sinus rhythm developing POAF [31]. This implies that microbiome products might play a significant role in the pathogenesis of POAF, but a strong causal relationship was not investigated within the study design [31]. In the present study, we analyzed the most extensively studied and well-known microbiome product (TMAO) trying to highlight the causal relationship between microbiome and POAF. Considering the previously published work, the analysis of fecal sample requires dedicated funds and expertise, and is not available in every center; therefore, it might take some years before being introduced into clinical practice. Although they only analyzed patients undergoing surgery without cardiopulmonary bypass to minimize the alterations related to atrial manipulation, intestinal permeability is influenced by extracorporeal circulation [32,33] and their results should be confirmed in the broader scenario.

Pericardial fluid analysis is providing biochemical markers to stratify the risk of developing POAF [34]. Since the epicardium acts as a semi-permeable membrane, the composition of pericardial fluids reflects the properties of the extracellular matrix and interstitial space. High levels of interleukin-6, mitochondrial DNA and natriuretic peptides appear to be promising early markers for POAF, but this still requires expansive analysis compared to a “focused” ELISA approach on serum TMAO, which is easier to collect, store and analyze compared to pericardial fluid. The same reasoning applies to genomic markers for predicting POAF, such as microRNA (miRNA) and other epigenetic mediators [35]; despite the limited application and availability, genetic and epigenetic mediators might provide critical perspective into arrhythmogenic substrates responsible for POAF.

The microbiome also interacts with the patients’ immune system with an immunomodulatory role in graft rejection after heart transplantation [36]. In patients with end-stage heart failure undergoing left ventricular assist device (LVAD) implantation or heart transplantation, serum levels of TMAO are persistently elevated both during the acute heart failure, and after ventricular assist device implantation, and therefore are considered independent of other measures of inflammation, endotoxemia, oxidative stress, and intestinal dysbiosis [37,38]. Moreover, microbiome changes have been recently considered in heart transplant patients [39] and may be associated with a different immunosuppressive treatment [40]. High levels of TMAO and the activation of the carnitine-butyrobetaine-TMAO cellular pathway are associated with increased risk of post-cardiac transplant rejection in the immediate postoperative period and increased atherosclerotic burden in the long term [41].

These studies highlighted how crucial it is to analyze the microbiota and its metabolites in cardiac surgery to improve outcomes both in the short and long term [42]. Reducing TMAO levels represent a promising strategy to modulate cardiac hypertrophy and fibrosis, leading to an antiarrhythmic effect. In animal models, inhibition of TMAO using 3,3-dimethyl-1-butanol (DMB) reduced structural and electrical remodeling, due to increased afterload and heart failure [43,44]. In conclusion, low levels of TMAO may reduce the risk of atrial fibrillation, both in the general population and in post-cardiac surgery patients.

### Limitations

In addition to the literature’s limitations regarding single-center observational or retrospective studies [45], this study was unable to include genetic analysis of the intestinal microbiome from fecal material. Furthermore, the pathways explored through serum and histological tissue analysis require further investigation to identify the most relevant signal cascades. Also, other metabolites of intestinal microbiota should be investigated. Additionally, it is necessary to verify the results in patients undergoing mitral or tricuspid valve surgery as they have higher risk of postoperative arrhythmic complications due to multifactorial pathogenesis. Also, in the case of non-elective procedures, other sources of LPS should be investigated (such as oral sources from periodontitis). These analyses should allow for a more precise selection of potential therapeutic strategies for future research.

## 4. Materials and Methods

### 4.1. Study Design and Inclusion Criteria

This study is a single-center, observational, non-profit study. No aspect of the study has changed clinical or surgical practice, and no invasive analyses beyond standard care were required. Patients were included in the study if: (1) undergoing cardiac surgery with cardiopulmonary bypass and central atrio-caval cannulation; (2) preoperative sinus rhythm without a history of atrial fibrillation; (3) signed informed consent. Exclusion criteria were: (1) patients undergoing procedures on the mitral, tricuspid, interatrial septum, or interventricular septum, either alone or in combination; (2) patients undergoing off-pump procedures (e.g., off-pump bypass); (3) patients undergoing urgent or emergency procedures (EuroSCORE criteria); (4) patients undergoing redo cardiac surgery procedures. Study received Institutional Review Board approval on 18 August 2020 (protocol code “Micro-AFib”, 64/20-OSS, Ethical Committee, Università Campus Bio-Medico di Roma, Rome, Italy). Patients’ enrollment was performed from 4 September 2020 to 15 May 2022.

### 4.2. Endpoints

The primary endpoint of the study is the occurrence of POAF. For the assessment of the primary endpoint, continuous ECG monitoring was performed on the patient from surgery until discharge, as per routine. The occurrence of postoperative atrial fibrillation and the therapies required were recorded in the medical history of the patient as per clinical practice. Secondary endpoints include all postoperative complications noted in the clinical diary (such as thoracentesis, use of blood products) and the duration of postoperative hospital stay. No outpatient follow-up was planned, and the study was concluded when the patient was discharged.

### 4.3. Sample Size Calculation

Considering the study hypotheses and the expected incidence of postoperative atrial fibrillation (30–35%), 100 patients were included in the study. Specifically, a minimum sample size for univariate regression was initially calculated, considering an expected odds ratio of 3, with a 10% event rate in the variable group, and standard alpha and beta error values (according to Hsieh [46]); a minimum initial sample size of 75 patients was obtained. Subsequently, the sample size was adjusted considering a maximum number of 3 independent variables in regression (k) (according to Blackstone [47]), with a proportion of patients with the outcome of 30% (p) (expected value from analyzing the literature), applying “sample size = 10 × k/p” (according to Peduzzi [48]), resulting in a final sample size of 100.

### 4.4. Sample Collection and Analysis

For each patient, a fragment of approximately 1 cm^2^ was taken from the right atrium during the procedure of venous cannulation (double-stage cannula in the right atrium and inferior vena cava) for cardiopulmonary bypass. The sample was analyzed using histological examination (morphology, inflammation, angiogenesis, and fibrosis), and fibrosis was quantitatively analyzed.

In addition to routine blood tests required for clinical practice the evening before surgery, a blood sample was taken and, after centrifugation, it was aliquoted and stored at −80 °C for subsequent enzyme-linked immunosorbent assay (ELISA) analysis.

From paraffin-embedded tissue blocks, hematoxylin and eosin staining (for morphology, inflammation, and angiogenesis) and Masson’s staining (for fibrosis) were performed, with qualitative grading from 0 to 3 using evaluation scales used in clinical practice (such as the “Abramov scoring system” and assessments from the Committee of American Heart Association) [49].

To assess mRNA expression, and to quantify the degree of fibrosis through quantitative real-time polymerase chain reaction (qRT-PCR), RNA extraction from histological samples was performed using Trizol (manufacturer’s protocol). A standard RNA sample was transcribed to cDNA using the SuperScript VILO cDNA synthesis kit (Thermofisher) and amplified via real-time PCR. The following genes were analyzed using the assays:Collagen type 1 [COL1], assay Hs00164004_m1Collagen type 3 [COL3], assay Hs00943809_m1Fibronectin, assay Hs00365052_m1TGFb, assay Hs00998133_m1SMAD-2, assay Hs00998187_m1

The expression of the analyzed genes was normalized to the expression of the reference gene glyceraldehyde-3-phosphate dehydrogenase (GADPH) (assay Hs99999905_m1) using the ΔΔCt method.

ELISA analyses were performed using validated commercial kits according to the manufacturer’s instructions to analyzed:sP-selectin, a marker of platelet activation, Thermofisher BMS219-4Lipopolysaccharide (LPS), a marker of bacterial presence, Cusabio CSB-E09945hZonulin (ZNL), a marker of intestinal permeability, Cusabio CSB-EQ027649HUTGFb, a marker of fibrosis, Thermofisher BMS249-4TMAO, as the main metabolic product of the gut microbiota, MyBioSource MBS7269386

Spectrophotometer readings were performed as shown on the data sheet. Analyses were performed in duplicate, and the mean value was recorded; duplicate measurements always showed a <10% difference.

### 4.5. Statistical Analysis

Continuous variables, based on normality testing (Shapiro–Wilk test), are expressed as mean and standard deviation, or median and interquartile range (25–75°), and group comparisons were made using parametric tests (e.g., *t*-test) or non-parametric tests (e.g., Mann–Whitney test); variables are graphically represented with box plots or histograms. Data are provided for the overall population and thus they are divided based on the analyzed endpoint. Discrete variables are reported in contingency tables and analyzed using the Chi^2^ test. For regression analyses, logistic regression (*logit* or *probit*) was used considering the binary main endpoint (“postoperative atrial fibrillation”). Variables with *p* < 0.200 in univariate analysis are included in the multivariable model (called “multivariate analysis”). The model is optimized with a backward stepwise approach, analyzing each step with likelihood ratio test and Wald test until obtaining a minimal model. Model fitness is evaluated using c-statistic, Hosmer–Lemeshow test, Link test, and with the partial residual plot. Cut-off analyses for any continuous variables are based on the area under the curve (AUC) to generate binary variables. Odds ratios are presented with their 95% confidence interval. The threshold for statistical significance is set at two-tails *p* value < 0.05. The analysis was performed using STATA ver. 17 (personal license).

## 5. Conclusions

A deep correlation has been confirmed between myocardial fibrosis and high serum levels of TMAO, leading to an increased risk of POAF, as shown in this study about the role of the gut microbiota in POAF and myocardial fibrosis. The gut microbiota and its main metabolite (i.e., TMAO) are important elements in the pathogenesis of myocardial fibrosis, the pathophysiological substrate of postoperative atrial fibrillation (POAF). At the tissue level, patients developing atrial fibrillation in the postoperative period (despite of being in preoperative sinus rhythm) show increased gene expression of collagen-1, collagen-3, and fibronectin, with a greater degree of myocardial fibrosis histologically. The TGFb/SMAD-2 signaling pathway might play an important role in these mechanisms, and in the cardiothoracic setting, as it is expressed more both in serum and in atrial tissue. Levels of bacteria (lipopolysaccharide) and TMAO are significantly higher in patients who develop POAF, and high blood levels of TMAO (≥61.8 ng/mL) are a risk factor for POAF.

However, myocardial fibrosis and postoperative arrhythmias have multiple pathophysiological aspects, and the microbiota is not the only cause. Nonetheless, modulation of the gut microbiota or its metabolites could be used as an additional strategy to protect the myocardium against fibrotic reactions, thus reducing the onset of arrhythmias and their related negative consequences in the short and long term. As the results of this study showed, inhibition of TMAO production could represent a potential therapeutic target to reduce cardiac fibrosis and consequent arrhythmic complications in patients undergoing cardiac surgery. Microbiota-based preventive and therapeutic interventions appear promising for the near future.

## Figures and Tables

**Figure 1 ijms-25-06037-f001:**
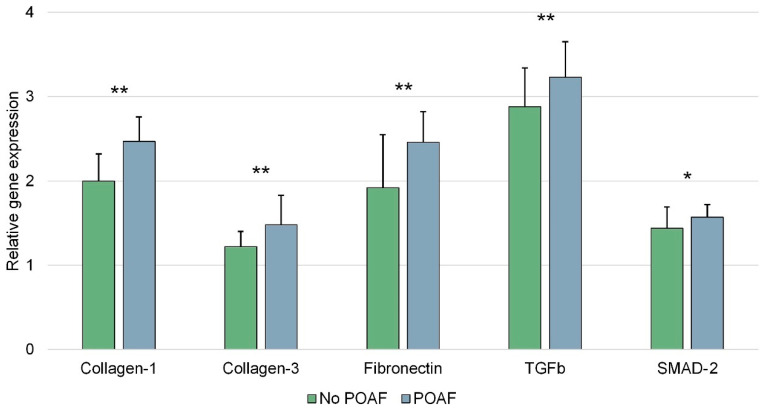
Relative gene expression (graphical visualization of data shown in Table 4). Statistically significant inter-group differences (POAF vs. No POAF) are marked as ** (if *p* < 0.01) or * (if *p* < 0.05) on top of the relevant column.

**Figure 2 ijms-25-06037-f002:**
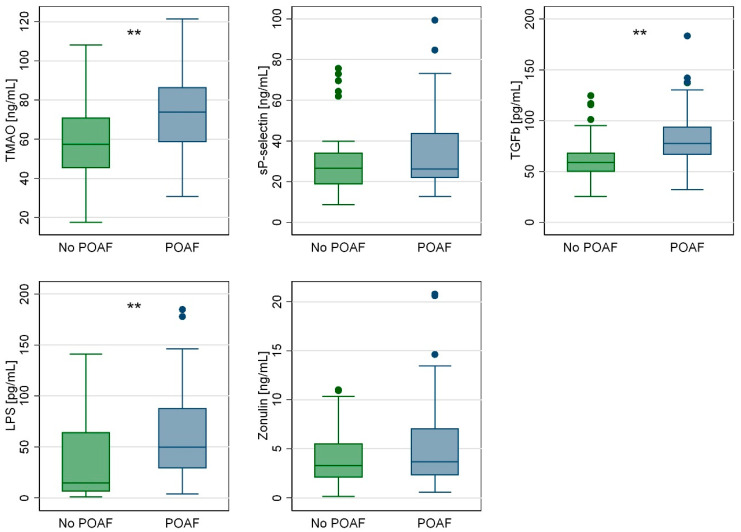
Box plot for ELISA analysis (graphical visualization of data shown in Table 5). Statistically significant inter-group differences (POAF vs. No POAF) are marked as ** (if *p* < 0.01) on top of the relevant box.

**Figure 3 ijms-25-06037-f003:**
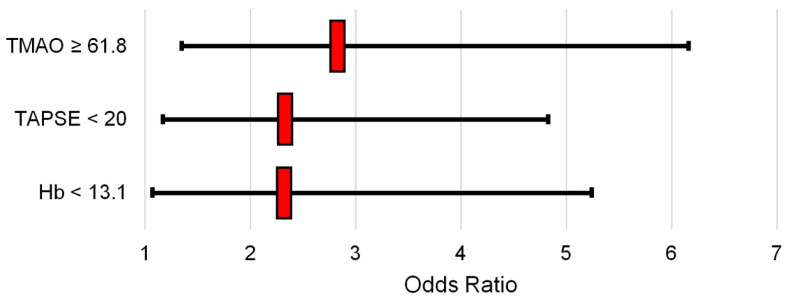
Dot plot of the odds ratio relative to the final model in Table 9.

**Table 1 ijms-25-06037-t001:** Baseline characteristics and intraoperative data.

	All Patients N = 100	POAF N = 38	No POAF N = 62	*p* Value
Age	69 (64–78)	68.5 (64–78)	69.5 (64–78)	0.646
Male sex	58 (58%)	24 (63.2%)	34 (54.8%)	0.413
Hypertension	98 (98%)	37 (97.4%)	60 (96.8%)	0.866
Dyslipidemia	75 (75%)	28 (73.7%)	47 (75.8%)	0.812
Diabetes	54 (54%)	24 (63.2%)	30 (48.4%)	0.150
Smoking habit	61 (61%)	29 (76.3%)	52 (83.9%)	0.350
Alcohol use (units/week)	4 (2–5)	3 (1–5)	4 (2–6)	0.102
Hemoglobin (g/dL)	12.8 (11.7–13.9)	12.2 (11.7–13.6)	13.1 (11.8–14.0)	0.062
Creatinine (mg/dL)	1.1 (0.9–1.3)	1.1 (0.8–1.3)	1.1 (0.9–1.3)	0.421
BMI (kg/m^2^)	31.9 (29.1–34.5)	31.9 (29.6–34.6)	31.3 (28.4–34.3)	0.432
LVEDD (mm)	51 (48–53)	51.5 (48–53)	49 (47–52)	0.098
LVEF (%)	55 (50–60)	55 (50–60)	50 (50–55)	0.081
LAVI	33 (31–35)	33 (31–35)	33 (30–35)	0.731
LAVI ≥ 34	44 (44%)	17 (44.7%)	27 (43.5%)	0.907
TAPSE (mm)	21 (17–22)	21 (17–23)	19.5 (17–22)	0.086
Surgery type:				0.857
Revascularization	82 (82%)	32 (84.2%)	50 (80.6%)	
Aortic valve	8 (8%)	3 (7.9%)	5 (8.1%)	
Ascending aorta	10 (10%)	3 (7.9%)	7 (11.3%)	
Duration CPB (min)	55 (41–69)	56 (45–71)	54 (40–69)	0.594
Duration XCL (min)	46 (30–58)	47 (32–59)	46 (30–58)	0.604
EuroSCORE II	2.21 (1.70–3.03)	2.27 (1.7–3.4)	2.20 (1.6–2.8)	0.454

CPB: cardiopulmonary bypass; XCL: aortic cross-clamp. BMI: body mass index; LVEDD: left ventricular end diastolic diameter; LVEF: left ventricular ejection fraction; LAVI: left atrial volume index, TAPSE: tricuspid annular plane systolic excursion.

**Table 2 ijms-25-06037-t002:** Postoperative complications.

	N = 100
Postoperative atrial fibrillation (POAF)	38 (38%)
Thoracentesis	26 (26%)
Postoperative length of stay (days)	7 (5.5–8)
Number of red packed blood cells (units)	1 (0–2)
In-hospital mortality	0 (0%)
Postoperative bleeding requiring mediastinal re-exploration	2 (2%)
Clinically relevant pericardial effusion (>1 cm or requiring drainage)	3 (3%)

**Table 3 ijms-25-06037-t003:** Histological evaluation of right atrium.

	All Patients N = 100	POAF N = 38	No POAF N = 62	*p* Value
Fibrosis				0.001
Grade 0	18 (18%)	0 (0%)	18 (29.0%)	
Grade 1	34 (34%)	6 (15.8%)	28 (45.2%)	
Grade 2	23 (23%)	12 (31.6%)	11 (17.7%)	
Grade 3	25 (25%)	20 (52.6%)	5 (8.1%)	
Fibrosis grade ≥ 2	48 (48%)	32 (84.2%)	16 (25.8%)	0.001
Angiogenesis				0.018
Grade 0–1	0 (0%)	0 (0%)	0 (0%)	
Grade 2	62 (62%)	18 (47.4%)	44 (71.0%)	
Grade 3	38 (38%)	20 (52.6%)	18 (29.0%)	
Inflammation				0.033
Grade 0	44 (44%)	14 (36.8%)	30 (48.4%)	
Grade 1	47 (47%)	17 (44.7%)	30 (48.4%)	
Grade 2	9 (9%)	7 (18.4%)	2 (3.2%)	

POAF: postoperative atrial fibrillation.

**Table 4 ijms-25-06037-t004:** Results of qRT-PCR for fibrosis.

	All Patients N = 100	POAF N = 38	No POAF N = 62	*p* Value
Collagen-1	2.2 (1.9–2.5) 2.18 ± 0.39	2.4 (2.2–2.7) 2.47 ± 0.29	2.0 (1.8–2.3) 2.00 ± 0.32	0.001
Collagen-3	1.3 (1.1–1.5) 1.32 ± 0.29	1.6 (1.1–1.8) 1.48 ± 0.35	1.2 (1.0–1.4) 1.22 ± 0.18	0.001
Fibronectin	2.2 (1.7–2.6) 2.13 ± 0.60	2.4 (2.1–2.8) 2.46 ± 0.36	2.0 (1.3–2.4) 1.92 ± 0.63	0.001
TGFb	3.0 (2.6–3.4) 2.98 ± 0.48	3.2 (2.9–3.6) 3.23 ± 0.42	2.9 (2.5–3.2) 2.88 ± 0.46	0.001
SMAD-2	1.5 (1.3–1.7) 1.49 ± 0.22	1.6 (1.5–1.7) 1.57 ± 0.15	1.5 (1.3–1.6) 1.44 ± 0.25	0.030

**Table 5 ijms-25-06037-t005:** Results of ELISA for POAF (primary endpoint).

	All Patients N = 100	POAF N = 38	No POAF N = 62	*p* Value
TMAO [ng/mL]	61.7 (51.6–75.2)	73.8 (58.6–86.7)	57.5 (45.2–71.1)	0.001
sP-selectin [ng/mL]	26.6 (20.0–36.5)	26.2 (21.7–44.0)	26.6 (18.6–34.2)	0.189
TGFb [pg/mL]	66.2 (51.5–75.8)	77.3 (66.2–94.1)	58.6 (49.6–68.5)	0.001
LPS [pg/mL]	28.9 (10.9–68.0)	49.7 (28.9–87.9)	14.7 (6.3–64.6)	0.001
Zonulin [ng/mL]	3.47 (2.13–5.94)	3.70 (2.30–7.08)	3.30 (2.08–5.56)	0.218

Values are expressed as median (interquartile range). See text for abbreviations.

**Table 6 ijms-25-06037-t006:** Regression analysis for POAF.

**Univariable Analysis**	**Odds Ratio**	**95% CI**	***p* Value**	**Included in Multivariable?**
Age	0.99	0.93–1.04	0.729	
Male sex	1.41	0.62–3.22	0.414	
Hypertension	1.23	0.11–14.08	0.866	
Dyslipidemia	0.89	0.35–2.25	0.812	
Diabetes	1.82	0.80–4.17	0.152	+
Smoking habit	0.62	0.22–1.69	0.352	
Alcohol use	0.79	0.65–0.97	0.028	+
Hemoglobin	0.77	0.54–1.08	0.138	+
Creatinine	0.50	0.09–2.64	0.418	
BMI	0.95	0.84–1.09	0.511	
LVEDD	0.89	0.78–1.02	0.104	+
LVEF	0.89	0.81–0.98	0.024	+
LAVI	0.96	0.82–1.13	0.643	
LAVI ≥ 34	1.04	0.46–2.36	0.907	
TAPSE	0.89	0.78–1.02	0.093	+
Surgery type				
Myocard. revasc	Ref.	Ref.	Ref.	
Aortic valve replac.	0.93	0.21–4.19	0.933	
Ascending aorta	0.67	0.16–2.78	0.581	
Duration CPB	1.01	0.98–1.03	0.629	
Duration XCL	1.01	0.98–1.03	0.618	
EuroSCORE II	1.28	0.78–2.10	0.322	
TMAO	1.05	1.02–1.08	0.001	+
sP-selectin	1.03	1.00–1.05	0.035	+
TGFb	1.04	1.02–1.06	0.001	+
LPS	1.01	1.00–1.02	0.003	+
Zonulin	1.10	0.99–1.23	0.059	+
**Multivariable model**	**Odds ratio**	**95% CI**		***p* value**
Hemoglobin	0.75	0.59–0.97		0.030
TAPSE	0.85	0.73–0.99		0.044
TMAO	1.05	1.02–1.08		0.001
TGFb	1.04	1.01–1.06		0.001

See other tables for abbreviations. Ref.: reference.

**Table 7 ijms-25-06037-t007:** Cut-off analysis for variables retained from multivariable model in Table 6.

Variable	AUC	95% CI	Cut-Off	Se	Sp	PPV	NPV
Hemoglobin	0.41	0.32–0.52	**13.1**	39.5	48.4	31.9	56.6
TAPSE	0.40	0.30–0.50	**20**	50.0	37.1	32.8	54.8
TMAO	0.77	0.68–0.85	**61.8**	71.1	85.5	73.5	80.3
TGFb	0.74	0.65–0.82	**72.1**	65.8	62.9	54.0	78.0

Se: sensibility, Sp: specificity; PPV: positive predictive value; NPV: negative predictive value; AUC: area under the receiver operating characteristic curve; CI: confidence interval.

**Table 8 ijms-25-06037-t008:** Multivariable regression model using binary variables, initial model.

	Odds Ratio	95% CI	*p* Value	Included in Final Model?
Hb ≥ 13.1 g/dL	0.40	0.17–0.97	0.042	+
TAPSE ≥ 20 mm	0.23	0.10–0.56	0.001	+
TMAO ≥ 61.8 ng/mL	7.18	2.57–20.03	0.001	+
TGFb ≥ 72.1 pg/mL	1.77	0.75–4.19	0.195	

Hb, preoperative hemoglobin; TAPSE: tricuspid annular plane systolic excursion (preoperative echocardiography); TMAO: trimethylamine-N-oxide (plasma); TGFb, Transforming Growth Factor beta (plasma), CI: confidence interval.

**Table 9 ijms-25-06037-t009:** Multivariable regression model using binary variables, final model.

	Odds Ratio	95% CI	*p* Value
Hb < 13.1 g/dL	2.37	1.07–5.24	0.033
TAPSE < 20 mm	2.38	1.17–4.83	0.017
TMAO ≥ 61.8 ng/mL	2.88	1.35–6.16	0.006

Hb, preoperative hemoglobin; TAPSE: tricuspid annular plane systolic excursion (preoperative echocardiography); TMAO: trimethylamine-N-oxide (plasma), CI: confidence interval.

**Table 10 ijms-25-06037-t010:** Supplementary analysis.

**Logit Regression with Interaction between TMAO and Fibrosis**			
	**Odds Ratio**	**95% CI**	***p* Value**
**Dependent variable: fibrosis (grade ≥ 2)**			
TMAO (≥61.8 ng/mL)	2.67	1.19–5.98	0.017
**Dependent variable: POAF (univariable)**			
TMAO (≥61.8 ng/mL)	4.16	1.74–9.93	0.001
Fibrosis (grade ≥ 2)	15.33	5.41–43.43	0.001
**Dependent variable: POAF (two variables)**			
TMAO (≥61.8 ng/mL)	3.43	1.23–9.58	0.019
Fibrosis (grade ≥ 2)	13.86	4.74–40.51	0.001
**Dependent variable: POAF (interaction)**			
Fibrosis # TMAO			
no yes	0.43	0.12–1.21	0.063
yes no	1.25	0.49–3.17	0.638
yes yes	2.75	1.22–6.17	0.014
**Dependent variable: POAF (factorial model)**			
Fibrosis (grade ≥ 2)	1.25	0.49–3.17	0.638
TMAO (≥61.8 ng/mL)	0.43	0.12–1.21	0.063
Fibrosis # TMAO			
yes yes	6.60	1.34–32.52	0.020
**Probit regression with endogenous covariates**			
	**Coefficient**	**95% CI**	***p* value**
**Dependent variable: POAF**			
Fibrosis (grade ≥ 2)	2.51	2.05–2.96	0.001
TMAO	2.08	1.75–2.1	0.001
**Dependent variable: fibrosis**			
TMAO	1.21	1.03–1.40	0.034
TGFb (ELISA)	1.51	1.32–1.71	0.018
/athrho2_1	−1.82	−2.85/−0.79	0.001
/lnsigma2	−0.74	−0.89/−0.61	0.001
Err. corr (e.fibrosis, e.POAF)	−0.95	−0.99/−0.66	-
SD (e.fibrosis)	0.47	0.41–0.54	-

## Data Availability

Data are available upon reasonable request to the corresponding author.

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
