# Peer review of "Intestinal Microbiota and Derived Metabolites in Myocardial Fibrosis and Postoperative Atrial Fibrillation"

_ijms, 2024, doi:10.3390/ijms25116037_

Round 1
Reviewer 1 Report
Comments and Suggestions for Authors
This is a very elegant and insightful study regarding the influence of intestinal microbiota and derived metabolites in myocardial fibrosis and postoperative atrial fibrillation. Indeed, this subject is interesting due to the lack of information and studies on the subject.
This is innovative, well-designed, and scientifically interesting.
Some minor changes should be made to increase manuscript quality:
1) Page 3, Line 102: At the end of the introduction the authors should say what is done in the study. Therefore section 2.1 should be transferred to the introduction “In this study the authors aimed at identifying risk factors associated with POAF by analyzing the role and effect of gut microbiota and its metabolites (such as TMAO) on 106 myocardial fibrosis (with histological study of the right atrium). Also, preoperative clinical or echocardiographic risk factors associated with POAF were identified.”
2) Page 3, Lines 110: Instead of “Study design” it should be “Study design and inclusion criteria”
3) Page 7, Line 241: Figure 1 caption should be clarified. Particularly the statistical signs must be clear. It is significant comparing what?
4) Page 8, Line 249: Same as above. Figure 2 caption should be clarified. It should be clear the type of comparison made and the meaning of “*”.
5) In the title it should be microbiota, not microbiote.
After these minor changes, the manuscript should be considered to be published.
Author Response
Reviewer 1
This is a very elegant and insightful study regarding the influence of intestinal microbiota and derived metabolites in myocardial fibrosis and postoperative atrial fibrillation. Indeed, this subject is interesting due to the lack of information and studies on the subject. This is innovative, well-designed, and scientifically interesting. Some minor changes should be made to increase manuscript quality. After these minor changes, the manuscript should be considered to be published.
Thank You for Your comments and Your appreciation. All Your points have been replied in the revised version of the manuscript. We hope You might find this version suitable for publication.
1) Page 3, Line 102: At the end of the introduction the authors should say what is done in the study. Therefore section 2.1 should be transferred to the introduction “In this study the authors aimed at identifying risk factors associated with POAF by analyzing the role and effect of gut microbiota and its metabolites (such as TMAO) on 106 myocardial fibrosis (with histological study of the right atrium). Also, preoperative clinical or echocardiographic risk factors associated with POAF were identified.”
Thank you. Changes have been made accordingly.
2) Page 3, Lines 110: Instead of “Study design” it should be “Study design and inclusion criteria”
Thank you. Changes have been made accordingly.
3) Page 7, Line 241: Figure 1 caption should be clarified. Particularly the statistical signs must be clear. It is significant comparing what?
Thank you. Figure 1 is a visual representation of data shown in Table 4. Caption has been revised accordingly.
4) Page 8, Line 249: Same as above. Figure 2 caption should be clarified. It should be clear the type of comparison made and the meaning of “*”.
Thank you. Figure 2 is a visual representation of data shown in Table 5. Caption has been revised accordingly.
5) In the title it should be microbiota, not microbiote.
Thank you, we apologize for the typo. Title has been changed accordingly.
Reviewer 2 Report
Comments and Suggestions for Authors
The study prepared by Antonio Nenna aims to analyze the microbiome's effect and its metabolites in the fibrosis of myocardial tissue in the right atrium, to explore its role in the pathogenesis of POAF.
As a general recommendation for the authors, the manuscript's Discussion section should be expanded with comparable results from the literature.
Other recommendations to the authors to improve the quality of the manuscript:
- The abstract should be a total of about 200 words maximum. It now includes more than 200 words.
- Kindly add a reference in line 58 according to the data presented. The same comment for the lines 154, 308.
- ’’The POAF is the most frequent complication, and it is observed in 38%, according to literature’s data’’. This sentence should be moved to the Discussion Section (lines 219-200). Please also add a reference regarding the ’’literature’s data’’.
- Kindly add a few microscopic images for the histological evaluation of the right atrium to illustrate the study results.
- In lines 303-304, ’’Several studies suggest that the gut microbiota might be closely linked to atrial fibrillation pathogenesis. [25].’’ but only one reference is listed.
- The authors should expand the mechanism that involves TMAO, as a metabolite derived from intestinal flora, that may be associated with the inflammatory response and the activation of pro-inflammatory signal related to atrial fibrillation in the Discussion Section. In this part of the manuscript, the authors should discuss and compare their results (point by point) with other studies available in the literature.
- Kindly revise the reference list according to the International Journal of Molecular Sciences journal recommendation https://www.mdpi.com/journal/ijms/instructions
Author Response
Reviewer 3
The study prepared by Antonio Nenna aims to analyze the microbiome's effect and its metabolites in the fibrosis of myocardial tissue in the right atrium, to explore its role in the pathogenesis of POAF.
- As a general recommendation for the authors, the manuscript's Discussion section should be expanded with comparable results from the literature.
Thank you for your comments and your appreciation. Discussion has been expanded accordingly.
- The abstract should be a total of about 200 words maximum. It now includes more than 200 words.
Thank you for your comment. Abstract has been shortened from about 450 words to about 250 words. It is extremely difficult to shorten the abstract further without losing relevant pieces of information. We hope that this does not prevent manuscript re-evaluation.
- Kindly add a reference in line 58 according to the data presented. The same comment for the lines 154, 308.
Thank you for your comment. References have been included accordingly.
- ’’The POAF is the most frequent complication, and it is observed in 38%, according to literature’s data’’. This sentence should be moved to the Discussion Section (lines 219-200). Please also add a reference regarding the ’’literature’s data’’.
Thank you for your comment. Changes have been made accordingly.
- Kindly add a few microscopic images for the histological evaluation of the right atrium to illustrate the study results.
Thank you for your comment. Histological images with different degrees of fibrosis have been included accordingly.
- In lines 303-304, ’’Several studies suggest that the gut microbiota might be closely linked to atrial fibrillation pathogenesis. [25].’’ but only one reference is listed.
Thank you for your comment. References have been included accordingly. Both articles are reviews that extensively discuss this topic, despite not in the setting of cardiac surgery.
- The authors should expand the mechanism that involves TMAO, as a metabolite derived from intestinal flora, that may be associated with the inflammatory response and the activation of pro-inflammatory signal related to atrial fibrillation in the Discussion Section. In this part of the manuscript, the authors should discuss and compare their results (point by point) with other studies available in the literature.
Thank you for your comment. Discussion has been expanded accordingly. A point-by-point comparison with available data remains difficult due to lack of other studies in this topic, as this field of research is still not adequately investigated due to limitations in funds and resources. A narrative comparison has been performed. We hope that this paper might foster the scientific debate and shed light on this topic.
- Kindly revise the reference list according to the International Journal of Molecular Sciences journal recommendation https://www.mdpi.com/journal/ijms/instructions.
Thank you for your comment. Reference formatting has been adjusted according to instruction for authors. DOIs have been included to facilitate tracking.
Reviewer 3 Report
Comments and Suggestions for Authors
I think the title of the article should be different. Indeed, the authors did not study the gut microbiome. They only studied TMAO (trimethylamine N-oxide) as a product of the microbiome. So, the title (as example) could be:
Association of trimethylamine N-oxide with myocardial fibrosis and post-operative atrial fibrillation.
Authors evaluated only one metabolite of intestinal microbiome. So, please check in the paper using plural form for this term (see, for example line 101).
Please, compress text in the Abstract. As part of the compression, remove p-values from the Abstract, but add abbreviations (TGFb, SMAD etc).
Introduction:
Add references about etiopathogenetic factors (lines 62-65).
Replace (through whole of the paper) capital letters by petit in full names of factors like Transforming Growth Factor, Mitogen-Activated Protein Kinase, Enzyme-Linked Immunosorbent Assay etc.
Add abbreviations for other terms, like SMAD, NLRP3 etc.
Please, add a several sentences at the end of the Introduction about the goal of the study.
The authors studied LPS but didn’t discuss possible sources of LPS in blood. Do they think LPS origin from the intestinal microbiome and excludes it because of other sources, like periodontitis? 61% of all studied in the paper patients are smoking, but tobacco use is the most important risk factor for periodontal disease.
There are problems with tables and text formatting starting from page 7, including font size (lines 243-247), missing titles of tables etc.
What's in lines 274-276?
Also see font size on lines 283–295.
Title of Table 7: What does the table title say? Cut-off analysis of what?
What do the black bars in the tables on pages 10 and 11 mean?
Please, use only initials for Author Contributions.
Author Response
Reviewer 2
I think the title of the article should be different. Indeed, the authors did not study the gut microbiome. They only studied TMAO (trimethylamine N-oxide) as a product of the microbiome. So, the title (as example) could be: “Association of trimethylamine N-oxide with myocardial fibrosis and post-operative atrial fibrillation”.
Thank you for your comments and your appreciation. Unfortunately, due to restrictions of our institution based on the PhD dissertation, the title cannot be radically changed. However, we revised the abstract and described this issue in limitations. We hope that You might find the revised version suitable for publication.
Authors evaluated only one metabolite of intestinal microbiome. So, please check in the paper using plural form for this term (see, for example line 101).
Thank you for your comments. Change have been made in abstract and text, accordingly.
Please, compress text in the Abstract. As part of the compression, remove p-values from the Abstract, but add abbreviations (TGFb, SMAD etc).
Thank you for your comment. Abstract has been shortened and abbreviations have been expanded.
Add references about etiopathogenetic factors (lines 62-65).
Thank you. References have been included accordingly.
Replace (through whole of the paper) capital letters by petit in full names of factors like Transforming Growth Factor, Mitogen-Activated Protein Kinase, Enzyme-Linked Immunosorbent Assay etc.
Thank you for your comment. Changes have been made accordingly.
Add abbreviations for other terms, like SMAD, NLRP3 etc.
Thank you for your comment. Changes have been made accordingly.
Please, add a several sentences at the end of the Introduction about the goal of the study.
Thank you. Changes have been made accordingly. The end of the Introduction and the first paragraph of Methods have been revised.
The authors studied LPS but didn’t discuss possible sources of LPS in blood. Do they think LPS origin from the intestinal microbiome and excludes it because of other sources, like periodontitis? 61% of all studied in the paper patients are smoking, but tobacco use is the most important risk factor for periodontal disease.
Thank you for your comment. We did not discuss oral sources of LPS in blood as all patients undergoing elective cardiac surgery have no signs / symptoms of oral / dental diseases. Therefore, periodontitis is always checked and cured before hospital admission, and surgical procedure is deferred to avoid detrimental complications. Urgent / emergent procedures were excluded by study design in this study. A sentence has been included in the discussion accordingly.
There are problems with tables and text formatting starting from page 7, including font size (lines 243-247), missing titles of tables etc. What's in lines 274-276? Also see font size on lines 283–295. What do the black bars in the tables on pages 10 and 11 mean?
Thank you for your comments. Problems with text and table formatting have been revised. Pagination has been revised accordingly.
Title of Table 7: What does the table title say? Cut-off analysis of what?
Thank you for your comment. Title has been revised accordingly. Cut-off analysis refers to variables retained from multivariable model in Table 6, to produce dichotomous variables that are clinically more useful compared to continuous variables.
Please, use only initials for Author Contributions.
Thank you. Changes have been made accordingly.
Round 2
Reviewer 2 Report
Comments and Suggestions for Authors
The quality of the manuscript has greatly improved with more details and clarifications. I recommend the paper for publication in its present form.